# Synergizing Motion and Appearance: Multi-Scale Compensatory Codebooks for Talking Head Video Generation

## Abstract

Talking head video generation aims to generate a realistic talking head video that preserves the person's identity from a source image and the motion from a driving video. Despite the promising progress made in the field, it remains a challenging and critical problem to generate videos with accurate poses and fine-grained facial details simultaneously. Essentially, facial motion is often highly complex to model precisely, and the one-shot source face image cannot provide sufficient appearance guidance during generation due to dynamic pose changes. To tackle the problem, we propose to jointly learn motion and appearance codebooks and perform multi-scale codebook compensation to effectively refine both the facial motion conditions and appearance features for talking face image decoding. Specifically, the designed multi-scale motion and appearance codebooks are learned simultaneously in a unified framework to store representative global facial motion flow and appearance patterns. Then, we present a novel multi-scale motion and appearance compensation module, which utilizes a transformer-based codebook retrieval strategy to query complementary information from the two codebooks for joint motion and appearance compensation. The entire process produces motion flows of greater flexibility and appearance features with fewer distortions across different scales, resulting in a high-quality talking head video generation framework. Extensive experiments on various benchmarks validate the effectiveness of our approach and demonstrate superior generation results from both qualitative and quantitative perspectives when compared to state-of-the-art competitors.

## 1 Introduction

Given a source image and a driving video, talking head video generation (Hong et al., 2022; Tao et al., 2024) aims to animate the person in the source image using the pose and expression from the driving video. Due to its widespread applications, such as video conferencing, the film industry, and virtual reality, it has attracted growing interest in the community.

Significant progress has been made on this task in terms of both quality and robustness in recent years. Existing works primarily focus on learning more accurate motion estimation and representation in 2D or 3D to enhance generation quality. Along the direction, unsupervised methods target predicting local motion flows around unsupervised keypoints without relying on facial priors (Siarohin et al., 2019b; Zhao & Zhang, 2022; Wang et al., 2024), and methods based on predefined models (e.g., 3DMM) (Zakharov et al., 2019; Zhang et al., 2023; Ha et al., 2020) focus on learning robust decoding features to generate high-quality face outputs. Despite the promising achievements, critical challenges persist: 1) Some motion patterns cannot be inferred from a single image pair solely relying on unsupervised keypoints or predefined models for motion estimation, as such models often have limited power of motion representation and may fail to capture certain dynamic aspects of the facial motion from single image pairs. 2) Even with accurate motion estimation, highly dynamic and complex motions in driving videos can create ambiguity during generation, as a still source image lacks sufficient appearance information to handle occluded regions or subtle expression changes. This results in noticeable artifacts and a significant drop in the quality of the generated output. Therefore, generating realistic-looking facial images not only requires inferring accurate motion flow between given two facial images but also needs to compensate for the intermediate appearance decoding feature from the one-shot source image for the final generation of face images.

In this work, we aim to synergize motion and appearance by simultaneously learning accurate motion flows for facial warping and robust facial appearance features for face image decoding, to ad-

vance talking head generation. We propose a unified framework that can achieve joint learning of both motion and appearance codebooks with multi-scale compensation. Specifically, to estimate the motion flow between two facial images (*i.e.*, source and driving), we design a multi-scale motion codebook that captures diverse motion patterns across scales from the entire dataset during training. Using this learned multi-scale motion codebook, we further devise a transformer-based compensation structure to iteratively refine motion flows in a coarse to fine manner. To enhance intermediate warped facial feature maps for image decoding, we introduce a multi-scale appearance codebook that represents diverse appearance patterns learned from the entire dataset. Using the learned appearance codebook, we introduce a transformer-based compensation structure to refine the warped features across different scales. This approach enables us to capture more facial details by leveraging the diverse appearance information contained within the codebook. To enhance the learning of both codebooks, we propose a joint training strategy in which the motion and appearance codebooks are learned simultaneously with the entire framework. This approach allows both codebooks to be optimized together, utilizing gradients from the refined warped features to strengthen their mutual influence and improve overall performance. By learning both multi-scale motion and appearance codebooks, our framework refines the motion flow to accurately warp the source facial features, which are further compensated with additional details from the appearance codebook. This process yields robust intermediate facial decoding features, resulting in improved generation.

We conduct extensive ablation studies to verify the effectiveness of the learned multi-scale motion and appearance codebooks. Experimental results demonstrate that both codebooks effectively enhance the motion flow and intermediate warped features, resulting in more accurate and detailed facial motion flows and feature textures. Furthermore, results on two challenging datasets indicate that our method surpasses state-of-the-art approaches, producing realistic-looking talking head videos. In summary, our contributions are threefold:

- We propose a novel framework that *jointly learns multi-scale motion and appearance codebooks*. The motion codebook captures motion patterns at varying levels of granularity, while the appearance codebook stores representative facial structure and texture features. This joint learning enables the model to effectively compensate both motion and appearance for advanced generation.

- We develop an effective *multi-scale compensation mechanism* that utilizes the learned motion and appearance codebooks to progressively refine both motion and appearance representations. The mechanism can couple the compensation of both aspects at each level, achieving higher consistency of appearance and motion, thus leading to high visual quality in generated videos.

- Extensive experiments including on challenging datasets demonstrate that our method not only effectively compensates for facial motions and appearances but also significantly outperforms state-of-the-art approaches, producing more realistic and visually convincing talking head videos.

## 2 RELATED WORK

**Talking Head Video Generation.** Existing works on talking head video generation generally separate the motion estimation module and image generation module to disentangle appearance and motion. To transfer the motion, some works require facial priors provided by a pre-trained model during generation. For example, landmark-based approaches (Ha et al., 2020; Zhao et al., 2021; Wu et al., 2018; Zakharov et al., 2020) detect pre-defined facial landmarks to transfer the facial pose and expression from a driving frame to the source image. Some other methods (Ren et al., 2021; Zeng et al., 2023; Yao et al., 2020) use parameters from 3D face models (Blanz & Vetter, 2023; Feng et al., 2021; Zhu et al., 2017) as motion descriptors to disentangle identity and pose. However, they normally cannot describe non-facial parts such as hair and neck, and their generation quality is limited by the pre-trained model performance. To address the issue, several methods that do not require any prior knowledge from pre-trained models are proposed. Monkey-Net (Siarohin et al., 2019a) learns sparse motion-related keypoints in an unsupervised manner to describe object movements. FOMM (Siarohin et al., 2019b) extends it with local affine transformation assumption around the keypoints to model complex motion. Subsequent works introduce more flexible mathematical models such as thin-plane spline transformation (Zhao & Zhang, 2022) and continuous piecewise-affine-based transformation (Wang et al., 2024) to increase motion estimation accuracy. Despite the expressiveness of the motion models, they cannot fully describe large head poses and delicate expression changes. MRFA (Tao et al., 2024) tackles the problem by building a correlation volume for each image pair and using it to refine the coarse motion flow iteratively. However, it only uses the warped image feature and a plain image generator for image generation, which may fail when facing extreme pose change as the appearance information from the one-shot source im-

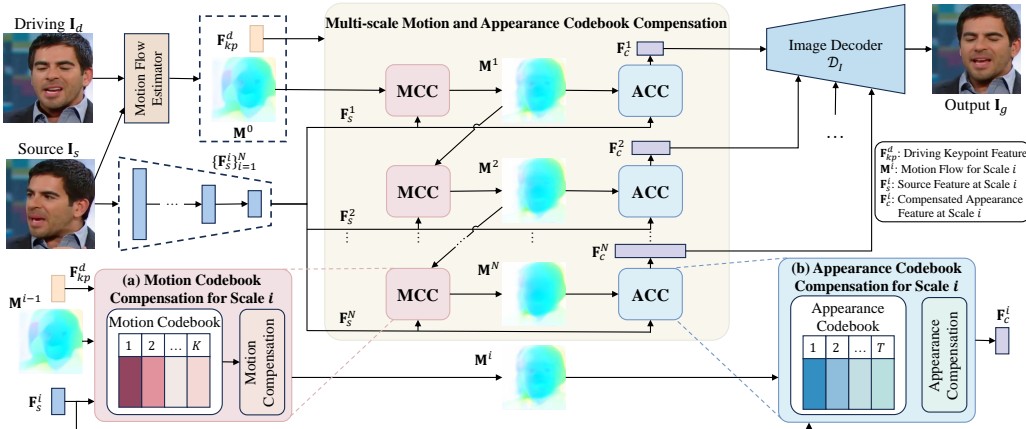

Figure 1: Overview of the framework. For each scale, it consists of two sub-modules. (i) **Motion Codebook Compensation (MCC)** compensates for a motion flow with the motion codebook. (ii) To compensate for the source facial feature warped by the compensated motion flow, **Appearance Codebook Compensation (ACC)** uses the appearance codebook and produces the compensated appearance feature for generation. These two sub-modules are employed for all scales. We learn the motion and appearance codebook jointly with the whole framework.

age is not enough. Some works (Yin et al., 2022; Oorloff & Yacoob, 2023; Bounareli et al., 2023) leverage the remarkable generative power of pre-trained StyleGAN (Karras et al., 2019; 2020) for better image generation, but they usually have to balance the editability and fidelity. MCNet (Hong & Xu, 2023) learns a meta-memory bank of spatial facial features to compensate for the warped source features. Different from previous works, we compensate for both motion flows and warped source features with jointly learned multi-scale motion and appearance codebooks to boost the final generation quality.

**Codebook Learning.** Codebook learning aims to learn useful discrete representations with a fixed size. The learned codebook contains rich and compact information, which can facilitate various tasks such as image classification (Cai et al., 2010; Zhang et al., 2009), image synthesis (Esser et al., 2021; Chang et al., 2022), blind face restoration (Zhou et al., 2022; Gu et al., 2022) and audio-driven talking head video generation (Wang et al., 2023). Traditional methods often learn a codebook with unsupervised clustering such as k-means (Csurka et al., 2004). VQ-VAE (Van Den Oord et al., 2017) first incorporates vector quantization in a Variational Autoencoder to learn a codebook containing discrete representative input features. To build a context-rich codebook for images, VQGAN (Esser et al., 2021) further increases the compression rate and adds a discriminator and a perceptual loss on images reconstructed with the codes from the codebook. A transformer is later used to model the composition of the codes for high-resolution image synthesis. CodeFormer (Zhou et al., 2022) uses a learned codebook of compressed high-quality face image features as discrete prior and predicts the code sequence based on the low-quality facial input for blind face restoration. LipFormer (Wang et al., 2023) learns two codebooks of the upper half face and the bottom half face respectively and predicts the lip codes from the input audio to generate a face video from the audio. We also adopt the idea of codebook learning, but we simultaneously learn multi-scale motion and appearance codebooks that store diverse motion and appearance patterns from the entire dataset during training to facilitate high-quality talking head video generation. The codebooks and the entire framework are trained together so that patterns useful for talking head video generation can be stored in and retrieved from the codebooks.

## 3 METHOD

In this section, we will present the details of our framework. We learn multi-scale motion and appearance codebooks with compensation for the motion and intermediate appearance features during generation. We adopt the Taylor expansion approximation method to learn the initial motion flow and the warping manner as same as Siarohin et al. (2019a) for video generation.

### 3.1 OVERVIEW

Our framework is illustrated in Fig. 1. First, a keypoint-based motion flow estimator takes both the source image $\mathbf{I}_s$ and driving image $\mathbf{I}_d$ as input and estimates the initial coarse motion flow $\mathbf{M}^0$. An

Figure 2: Illustration of motion codebook learning and compensation for scale $i$. We adopt a transformer structure $\mathcal{T}_M$ to compensate for the motion flow using the learned motion codebook. The proposed motion codebook is learned under the supervision of a reconstruction loss and a code-level loss.

image encoder $\mathcal{E}_I$ extracts multi-scale source features $\{\mathbf{F}_s^i\}_{i=1}^N$ from $\mathbf{I}_s$. Using $\mathbf{M}^0$, $\{\mathbf{F}_s^i\}_{i=1}^N$, and the driving keypoint feature $\mathbf{F}_{kp}^d$, the multi-scale motion and appearance codebook compensation module refines the motion flow and warped source features across all scales. At each scale, motion codebook compensation refines the motion flow $\mathbf{M}^{i-1}$, and the compensated motion flow $\mathbf{M}^i$ is used to warp the source feature $\mathbf{F}_s^i$. Appearance codebook compensation then refines the warped source feature to produce the compensated appearance feature $\mathbf{F}_c^i$. If $i < N$, $\mathbf{M}^i$ is used as the motion flow for the next scale. Finally, the image decoder $\mathcal{D}_I$ decodes the compensated appearance features $\{\mathbf{F}_c^i\}_{i=1}^N$ to generate the final image $\mathbf{I}_g$ with the target motion and appearance. Details on the design and learning of multi-scale motion and appearance codebooks and compensation are provided in the following subsections.

## 3.2 Multi-scale Motion Codebook Learning and Compensation

Previous methods typically estimate motion flows with source and driving features from a fixed scale with certain mathematical models. While they capture rough motion, their accuracy is limited by the single-scale information and the assumed transformations around unsupervised keypoints, especially in complex motion scenarios. To improve this, we refine the initial motion flow $\mathbf{M}^0$ from coarse to fine using a multi-scale motion codebook. This enables us to produce more accurate motion flows for warping source features at different scales. Specifically, we learn a multi-scale motion codebook that stores local motion flow patterns and retrieve relevant information from it. Fig. 2 illustrates the motion codebook learning and compensation process for scale $i$.

**Multi-scale Motion Code Allocation.** We refine the initial motion flow using a multi-scale motion codebook across all $N$ scales. As the scale increases, larger source features require finer motion flows for accurate warping. To provide detailed motion flow compensation at larger scales, we introduce a code allocation scheme as shown in Fig. 3 for the multi-scale motion codebook, which divides the codebook into multiple groups and allocates more codes for larger scales. Specifically, the motion codebook $\mathcal{C}_M = \{\mathbf{m}_k \in \mathbb{R}^{d_m}\}_{k=1}^K$ contains $K$ codes, and we perform motion compensation at $N$ different scales. The $K$ codes are split into $N$ equal groups, each with $K/N$ codes. At scale $i$, the first $i$ groups, totaling $N_m^i = i \times K/N$ codes, are allocated for motion compensation. This allows codes with smaller indices to capture general motion patterns shared across scales, while codes with larger indices to capture finer motion patterns needed for larger scales. The code allocation scheme maximizes the use of the motion codebook by sharing general motion information across scales while reserving space for scale-specific details. At each scale $i$, we form a new motion codebook $\mathcal{C}_M^i$ from the allocated codes, resulting in $N$ scale-specific motion codebooks $\{\mathcal{C}_M^i\}_{i=1}^N$ for multi-scale motion compensation.

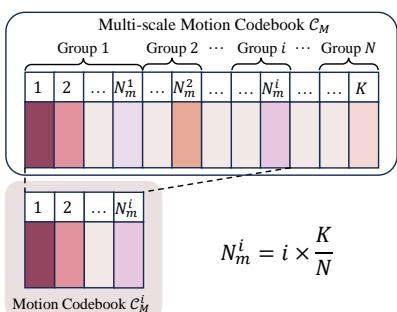

Figure 3: Illustration of the code allocation scheme. We take the multi-scale motion codebook as an example.

**Multi-scale Motion Codebook Learning.** To learn a motion codebook storing multi-scale motion flow patterns, we update $\{\mathcal{C}_M^i\}_{i=1}^N$ following the idea of vector quantization (Van Den Oord et al.,

2017). At scale $i$, a CNN-based motion flow encoder $\mathcal{E}_M$ maps the input motion flow $\mathbf{M}^{i-1} \in \mathbb{R}^{h \times w \times 2}$ into a compact motion flow feature $\mathbf{F}_m^{i-1} \in \mathbb{R}^{h_m \times w_m \times d_m}$, in which each unit is of length $d_m$ and captures a local motion flow pattern on $\mathbf{M}^{i-1}$. We quantize each of its spatial element $\mathbf{F}_m^{i-1}(x, y)$ with its nearest code from $\mathcal{C}_M^i$ to obtain a quantized motion flow feature $\hat{\mathbf{F}}_m^{i-1}$:

$$\hat{\mathbf{F}}_m^{i-1} = Q(\mathbf{F}_m^{i-1}) := \left( \underset{\mathbf{m}_k \in \mathcal{C}_M^i}{\arg\min} \, ||\mathbf{F}_m^{i-1}(x, y) - \mathbf{m}_k||_2^2 \right) \in \mathbb{R}^{h_m \times w_m \times d_m}. \tag{1}$$

A motion flow decoder $\mathcal{D}_M$ reconstructs $\mathbf{M}^{i-1}$ with the quantized motion flow feature $\hat{\mathbf{F}}_m^{i-1}$:

$$\hat{\mathbf{M}}^{i-1} = \mathcal{D}_M(\hat{\mathbf{F}}_m^{i-1}) = \mathcal{D}_M(Q(\mathcal{E}_M(\mathbf{M}^{i-1}))). \tag{2}$$

To update the scale-specific motion codebook $\mathcal{C}_M^i$ with local motion flow patterns from $\mathbf{F}_m^{i-1}$, we use the following loss function:

$$\mathcal{L}_{vq,m}^i = \lambda_{recon,m} ||\hat{\mathbf{M}}^{i-1} - sg[\mathbf{M}^{i-1}]||_1 + ||sg[E_M(\mathbf{M}^{i-1})] - \hat{\mathbf{F}}_m^{i-1}||_2^2$$
$$+ \beta ||sg[\hat{\mathbf{F}}_m^{i-1}] - E_M(sg[\mathbf{M}^{i-1}])||_2^2, \tag{3}$$

where $sg[\cdot]$ denotes the stop gradient operator, and $\lambda_{recon,m}$ and $\beta$ are the weights of the loss terms. The first term represents the motion flow reconstruction loss, while the last two terms form a code-level loss (Van Den Oord et al., 2017) that minimizes the distance between the latent motion flow units and the motion codes. We stop the gradient of $\mathbf{M}^{i-1}$ to ensure that motion codebook learning at scale $i$ does not interfere with the training of the motion flow estimator or the compensated motion flow from prior scales. The overall loss function for multi-scale motion codebook learning across all $N$ scales is $\mathcal{L}_{vq,m} = \sum_{i=1}^N \mathcal{L}_{vq,m}^i$.

**Multi-scale Motion Codebook Compensation.** An intuitive approach for multi-scale motion code-book compensation is to retrieve motion codes from the scale-specific codebook and decode them into finer motion flows using $\mathcal{D}_M$ at each scale. However, to reduce computational complexity, we limit the number of motion codes, which decreases expressiveness. This makes it difficult to fully reconstruct motion flows, leading to degraded image quality. Instead, we predict the motion flow residual for each input, allowing the motion codebook to enhance the flow without the need for precise reconstruction.

As shown in Fig. 2, to retrieve motion residuals from $\mathcal{C}_M^i$ at scale $i$, we use a motion code retrieval transformer $\mathcal{T}_M$ shared for all scales. It queries the scale-specific codebook $\mathcal{C}_M^i$ using the encoded motion feature $\mathbf{F}_m^{i-1}$, the warped source feature $\mathbf{F}_{cw}^{i-1}$, and the driving keypoint feature $\mathbf{F}_{kp}^d$. The first two represent the current motion flow, and the last indicates the target pose. These features are processed through a convolutional encoding block and concatenated into a compact motion query feature, then flattened and enhanced with a learnable position embedding before being passed to $\mathcal{T}_M$. $\mathcal{T}_M$ consists of $L_M$ transformer layers, each with multi-head self-attention, cross-attention, and convolution layers (instead of linear layers) to preserve spatial structure. The self-attention models global correlations, and the cross-attention uses the output of self-attention as the query and the codes from $\mathcal{C}_M^i$ as key-value pairs to retrieve local motion flow patterns. The transformer outputs the motion flow residual feature $\mathbf{F}_{mr}^{i-1}$, which is decoded by a motion flow residual decoder $\mathcal{D}_{MR}$ to obtain the motion flow residual $\mathbf{M}_r^i \in \mathbb{R}^{h \times w \times 2}$, Finally, we add $\mathbf{M}_r^i$ to the input motion flow $\mathbf{M}^{i-1}$ to get the compensated motion flow $\mathbf{M}^i$, which is used for source feature warping at scale $i$.

The compensated motion flow $\mathbf{M}^i$ is sufficient for warping the source feature $\mathbf{F}_s^i$, but may lack fine details for higher-resolution features like $\mathbf{F}_s^{i+1}$. Therefore, we use $\mathbf{M}^i$ as input for motion codebook compensation at scale $i+1$ and refine it with more motion codes. This iterative process continues across scales, producing multi-scale compensated motion flows $\{\mathbf{M}^i\}_{i=1}^N$.

### 3.3 Multi-scale Appearance Codebook Learning and Compensation

We have obtained compensated motion flows for each scale, allowing us to warp the source features more accurately. However, when the motion between source and driving images is too large, occlusion can cause the warped features to lack sufficient appearance details. To address this, we use appearance codebook compensation at each scale. Specifically, we learn a multi-scale appearance codebook with local textures and retrieve appropriate appearance information to enhance the warped features for image generation. Fig. 4 illustrates the appearance codebook learning and compensation at scale $i$.

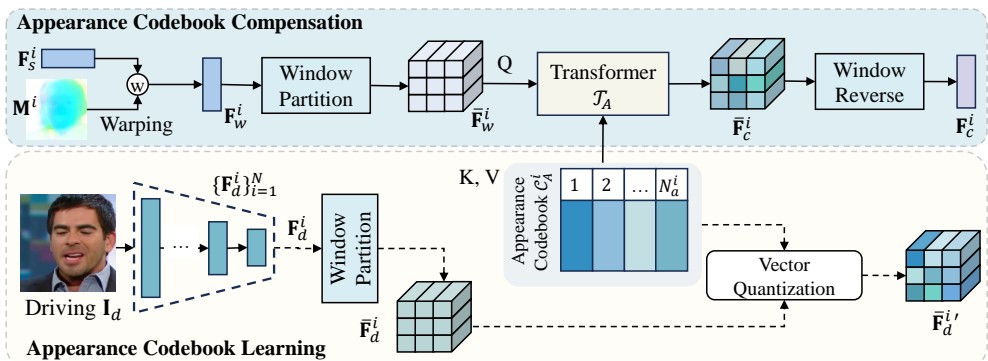

Figure 4: Illustration of appearance codebook learning and compensation for scale $i$. We utilize a transformer structure $\mathcal{T}_A$ to compensate for the warped features, adding more facial details to the feature map. Similar to the motion codebook, the designed appearance codebook is learned under the supervision of a code-level loss.

**Multi-scale Appearance Code Allocation.** We aim to compensate for the warped source features across all $N$ scales. As the scale increases, larger features require more detailed appearance information for compensation. To achieve this, we also perform the code allocation scheme in Fig. 3 on the multi-scale appearance codebook. The appearance codebook $\mathcal{C}_A = \{\mathbf{a}_k \in \mathbb{R}^{d_a}\}_{k=1}^T$, containing $T$ codes, is divided into $N$ groups, each with $T/N$ codes. At scale $i$, we allocate the first $i$ groups, which are the first $N_a^i = i \times T/N$ codes for appearance compensation. Codes with smaller indices capture general, coarse appearance patterns shared across scales, while larger-index codes focus on finer details. We form a new appearance codebook $\mathcal{C}_A^i$ for each scale $i$, resulting in $N$ scale-specific appearance codebooks $\{\mathcal{C}_A^i\}_{i=1}^N$ for multi-scale compensation.

**Multi-scale Appearance Codebook Learning.** To compensate for multi-scale warped source features, we aim to store appearance codes of different scales in the appearance codebook. Similar to the multi-scale motion codebook, we use vector quantization to directly update $\{\mathcal{C}_A^i\}_{i=1}^N$ for the multi-scale appearance codebook learning. We extract multi-scale driving features $\{\mathbf{F}_d^i\}_{i=1}^N$ using the image encoder $\mathcal{E}_I$, which serve as targets for our appearance codebook. Since image features of different scales have varying resolutions, directly flattening high-resolution features can be computationally expensive. To address this, we apply window partitioning. For a feature map of shape $(h_a^i, w_a^i, c_a^i)$ at scale $i$, we divide it into patches of shape $(h_a^i/h_a, w_a^i/w_a)$, reshaping the feature into $(h_a, w_a, c_a^i \times h_a^i \times w_a^i/h_a/w_a)$. We then linearly project the features into $d_a$ dimensions to align them with the appearance codes. This results in a more compact driving feature $\overline{\mathbf{F}}_d^i \in \mathbb{R}^{h_a \times w_a \times d_a}$, where each unit represents an appearance pattern at scale $i$. Finally, element-wise quantization with the nearest code from $\mathcal{C}_A^i$ produces the quantized appearance feature $\overline{\mathbf{F}}_d^{i\prime}$.

$$\overline{\mathbf{F}}_d^{i\prime} = Q(\overline{\mathbf{F}}_d^i) := \left( \operatorname*{arg\,min}_{\mathbf{a}_k \in \mathcal{C}_A^i} ||\overline{\mathbf{F}}_d^i(x,y) - \mathbf{a}_k||_2^2 \right) \in \mathbb{R}^{h_a \times w_a \times d_a}. \tag{4}$$

To update the scale-specific appearance codebook $\mathcal{C}_A^i$ with local appearance units from $\overline{\mathbf{F}}_d^i$, we use the following loss function:

$$\mathcal{L}_{vq,a}^i = ||sg[\overline{\mathbf{F}}_d^i] - \overline{\mathbf{F}}_d^{i\prime}||_2^2 + \beta ||sg[\overline{\mathbf{F}}_d^{i\prime}] - \overline{\mathbf{F}}_d^i||_2^2, \tag{5}$$

where $sg[\cdot]$ denotes the stop gradient operator, and $\beta$ is the weight of the loss term. We only use the code-level loss to reduce the distance between the appearance feature units and the codes from $\mathcal{C}_A^i$. The overall loss function for multi-scale appearance codebook learning is $\mathcal{L}_{vq,a} = \sum_{i=1}^N \mathcal{L}_{vq,a}^i$. We do not use the image decoder $\mathcal{D}_I$ to reconstruct the original driving image $\mathbf{I}_d$ from $\{\overline{\mathbf{F}}_d^{i\prime}\}_{i=1}^N$, allowing $\mathcal{D}_I$ to focus on decoding the compensated appearance features and improving the talking head video generation. Training a separate image decoder for reconstruction with quantized appearance codes would be computationally expensive. Experimental results in Sec. 4.3 show that using only the code-level loss effectively learns the appearance codebook.

**Multi-scale Appearance Codebook Compensation.** To transfer the compensated motion to the source image, we warp the source features with their corresponding compensated motion flows at

each scale. However, warping can introduce distortions due to pose changes, which degrade image quality. To address this, we use the multi-scale appearance codebook to repair the distorted warped features with retrieved appearance information. At scale $i$, we first warp the source feature $\mathbf{F}_s^i$ with the compensated motion flow $\mathbf{M}^i$ to obtain the warped feature $\mathbf{F}_w^i$. We then apply window partitioning to map $\mathbf{F}_w^i$ into a compact feature $\overline{\mathbf{F}}_w^i$. To correct the corrupted appearance in $\overline{\mathbf{F}}_w^i$, we retrieve appearance codes with $\overline{\mathbf{F}}_w^i$ using a transformer $\mathcal{T}_A$. $\overline{\mathbf{F}}_w^i$, reshaped and augmented with a learnable position embedding, passes through $L_A$ transformer layers, which also use convolution layers instead of linear layers to preserve spatial structures. The self-attention mechanism models global interactions, and cross-attention retrieves appearance codes from $\mathcal{C}_A^i$. The transformer output is the compensated appearance feature $\overline{\mathbf{F}}_c^i$, which retains the pose but reduces distortion of $\overline{\mathbf{F}}_w^i$. Finally, we apply window reverse on $\overline{\mathbf{F}}_c^i$ to restore it to the original image shape $(h_a^i, w_a^i, c_a^i)$ using linear projection and reshaping. This appearance codebook compensation is performed across all $N$ scales, resulting in multi-scale compensated appearance features $\{\mathbf{F}_c^i\}_{i=1}^N$.

### 3.4 Joint Optimization Objectives of the Framework

We use the multi-scale compensated appearance features $\{\mathbf{F}_c^i\}_{i=1}^N$ and the image decoder $\mathcal{D}_I$ for image generation. The low-resolution feature $\mathbf{F}_c^1$ is fed as the initial input to $\mathcal{D}_I$, which gradually upsamples it through a series of upsampling layers and ResNet blocks (He et al., 2016) until it reaches the output resolution. For higher-resolution features $\{\mathbf{F}_c^i\}_{i=2}^N$, we apply SFT (Wang et al., 2018) to refine the intermediate features when they match the resolution of $\mathbf{F}_c^i$. We also pass $\mathbf{F}_c^i$ through a convolution layer and add the result to the intermediate features. After fusing all scales, $\mathcal{D}_I$ generates the final image $\mathbf{I}_g$.

While we introduced multi-scale motion and appearance codebooks separately in Sec. 3.2 and Sec. 3.3, unlike traditional codebook-based methods (Esser et al., 2021; Zhou et al., 2022; Xing et al., 2023) where codebooks are pre-trained, we train both codebooks and the full framework end-to-end. This allows the network to effectively store and retrieve useful patterns at different scales, leading to high-quality talking head video generation.

We follow the unsupervised training pipeline from (Siarohin et al., 2019b), where the source and driving frames are extracted from the same video, and our framework learns to reconstruct the driving frame. The training combines the codebook losses from Sec. 3.2 and Sec. 3.3 with common losses for talking head video generation (Siarohin et al., 2019b; Hong et al., 2022). Specifically, we use the equivariance loss $\mathcal{L}_{eq}$ and keypoint distance loss $\mathcal{L}_{kpd}$ to guide keypoint prediction in the motion flow estimator, and an image reconstruction loss $\mathcal{L}_{recon}$ and an adversarial loss $\mathcal{L}_{adv}$ on the output $\mathbf{I}_g$. To avoid the image decoder relying too much on high-resolution appearance features, we also generate an image $\mathbf{I}_g^1$ using only $\mathbf{F}_c^1$ and apply $\mathcal{L}_{recon}$ to minimize the difference between $\mathbf{I}_g^1$ and $\mathbf{I}_d$. The overall training objective is:

$$\mathcal{L} = \mathcal{L}_{recon}(\mathbf{I}_d, \mathbf{I}_g) + \lambda_{adv}\mathcal{L}_{adv}(\mathbf{I}_d, \mathbf{I}_g) + \mathcal{L}_{eq} + \mathcal{L}_{kpd} + \mathcal{L}_{vq,m} + \mathcal{L}_{vq,a} + \lambda^1\mathcal{L}_{recon}(\mathbf{I}_d, \mathbf{I}_g^1), \quad (6)$$

where $\lambda_{adv}$ and $\lambda^1$ are the weights of the corresponding loss terms.

## 4 Experiments

### 4.1 Implementation Details

**Datasets.** We conduct experiments on VoxCeleb1 (Nagrani et al., 2017) and CelebV-HQ (Zhu et al., 2022) datasets. We train our model on VoxCeleb1 training set. For evaluation, we build the test set on VoxCeleb1 by randomly sample 50 videos from its test split. To evaluate the model's generalization ability, we randomly select 50 videos from CelebV-HQ for testing.

**Metrics.** For same-identity reconstruction, we adopt PSNR, $\mathcal{L}_1$ and LPIPS following (Tao et al., 2024) to evaluate the reconstruction quality. We also use FID (Heusel et al., 2017) to measure the realism of the generated video frames. Following (Siarohin et al., 2019a), we employ Average Keypoint Distance (AKD) for motion transfer quality evaluation and Average Euclidean Distance (AED) for identity preservation quality evaluation.

### 4.2 Comparison with State-of-the-Art Methods

We compare our method with a series of open-source state-of-the-art methods including non-diffusion based FOMM (Siarohin et al., 2019b), DaGAN (Hong et al., 2022), TPSM (Zhao & Zhang, 2022), MCNet (Hong & Xu, 2023), MRFA (Tao et al., 2024) and LivePortrait (Guo et al., 2024), and diffusion-based AniPortrait (Wei et al., 2024) and Follow-Your-Emoji (FYE) (Ma et al., 2024).

Table 1: Quantitative comparison with state-of-the-art methods for same-identity reconstruction on VoxCeleb1 and CelebV-HQ dataset. Our results are the best on Voxceleb1 dataset and competitive on CelebV-HQ dataset.

| Method | VoxCeleb1 | | | | | | CelebV-HQ | | | | | |
|---|---|---|---|---|---|---|---|---|---|---|---|---|
| | FID ↓ | PSNR ↑ | $\mathcal{L}_1$ ↓ | LPIPS ↓ | AKD ↓ | AED ↓ | FID ↓ | PSNR ↑ | $\mathcal{L}_1$ ↓ | LPIPS ↓ | AKD ↓ | AED ↓ |
| FOMM Siarohin et al. (2019b) | 53.97 | 22.96 | 0.0474 | 0.2200 | 1.4037 | 0.1509 | 78.15 | 20.92 | 0.0685 | 0.2925 | 3.6098 | 0.2955 |
| DaGAN Hong et al. (2022) | 51.55 | 22.92 | 0.0492 | 0.2251 | 1.5740 | 0.1652 | 99.84 | 20.49 | 0.0781 | 0.3209 | 7.6075 | 0.3301 |
| TPSM Zhao & Zhang (2022) | 53.69 | 24.73 | 0.0402 | 0.1974 | 1.2338 | 0.1241 | 73.14 | 22.19 | 0.0645 | 0.2618 | 4.5840 | 0.2843 |
| MCNet Hong & Xu (2023) | 51.45 | 24.59 | 0.0402 | 0.1996 | 1.2363 | 0.1254 | 78.33 | 22.20 | 0.0640 | 0.2732 | 4.1386 | 0.2903 |
| MRFA Tao et al. (2024) | 48.49 | 25.26 | 0.0370 | 0.1872 | **1.1823** | 0.1188 | 75.73 | 22.41 | 0.0625 | 0.2670 | 3.7166 | **0.2527** |
| AniPortrait Wei et al. (2024) | 52.65 | 20.15 | 0.0637 | 0.2767 | 2.6543 | 0.2623 | 66.67 | 17.37 | 0.1027 | 0.3263 | **1.9444** | 0.3517 |
| FYE Ma et al. (2024) | 43.25 | 19.54 | 0.07137 | 0.2954 | 2.7071 | 0.2652 | 62.55 | 19.58 | 0.0802 | 0.3006 | 4.8637 | 0.3029 |
| LivePortrait Guo et al. (2024) | 48.11 | 22.94 | 0.0484 | 0.2213 | 1.5516 | 0.1602 | **53.88** | 21.25 | 0.0659 | **0.2601** | 2.0467 | 0.2718 |
| Ours | **43.15** | **25.30** | **0.0355** | **0.1846** | 1.2039 | **0.1071** | 71.78 | 22.40 | **0.0610** | 0.2608 | 3.2562 | 0.2825 |

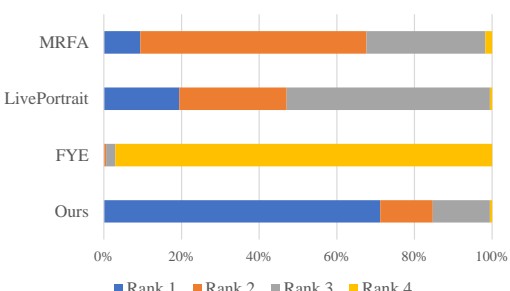

| Source | Driving | MRFA | LivePortrait | FYE | Ours | Source | Driving | MRFA | LivePortrait | FYE | Ours |

(a) Same-identity Reconstruction        (a) Cross-identity Reenactment

Figure 5: Qualitative comparison with state-of-the-art methods for (a) same-identity reconstruction and (b) cross-identity reenactment on VoxCeleb1 and CelebV-HQ dataset.

**Same-identity Reconstruction.** To evaluate the performance on same-identity reconstruction, we use the first frame of each video as the source image and reconstruct the whole video. Tab. 1 presents the quantitative results for same-identity reconstruction. Our method outperforms the other methods almost on all metrics on VoxCeleb1 dataset and remains competitive when generalized to the more challenging CelebV-HQ. Compared with those unsupervised methods (Siarohin et al., 2019b; Hong et al., 2022; Hong & Xu, 2023; Tao et al., 2024), our method achieves better motion estimation, *e.g.,*, our method get the best AKD scores on CelebV-HQ dataset. This result verifies the effectiveness of our designed multi-scale motion codebook and its generalizability. For the results of image quality (*i.e., FID, PSNR, $\mathcal{L}_1$, LPIPS*), our method outperforms other methods on VoxCeleb1 dataset, even those diffusion methods (Wei et al., 2024; Ma et al., 2024) and Guo et al. (2024) trained with larger-scale datasets. It indicates that our designed multi-scale appearance codebook is capable to compensate for the intermediate warped feature for better talking head video generation. We also show some qualitative results in Fig. 4.2 (a). Our method can generate plausible unseen facial regions (the second row and the third row) and eliminate the undesired occlusion (the fourth row) in the source image while preserving the motion faithfully.

**Cross-identity Reenactment.** We conduct cross-identity reenactment experiments to validate our method. Since there is no ground truth for this setting, we perform a user study comparing our approach to recent SOTA methods, including two GAN-based models (MRFA (Tao et al., 2024) and LivePortrait (Guo et al., 2024)) and one diffusion model (Follow-You-Emoji (FYE) (Ma et al., 2024)). We randomly selected 10 source-driving pairs and asked 30 participants to evaluate the generated videos based on appearance realism, motion naturalness, and overall quality. The results, shown in Fig. 6 and

Figure 6: User study results of ranking the quality of videos generated by different methods.

Fig. 4.2(b), indicate that users preferred our method. FYE (Ma et al., 2024) often produced exaggerated expressions and struggled to imitate the driving expressions accurately, likely due to its reliance on landmark-based embeddings without explicitly modeling motion. Compared to other GAN methods (Tao et al., 2024; Guo et al., 2024), our method better preserves facial shape and expression consistency. These findings confirm the effectiveness of our framework.

Table 2: Ablation study on the multi-scale motion and appearance codebook compensation. We present the results for same-identity reconstruction on VoxCeleb1 dataset.

| | FID ↓ | PSNR ↑ | $\mathcal{L}_1$ ↓ | LPIPS ↓ | AKD ↓ | AED ↓ |
|---|---|---|---|---|---|---|
| Baseline | 47.83 | 24.93 | 0.0375 | 0.1954 | 1.2384 | 0.1106 |
| Baseline + SMC | 49.00 | 24.97 | 0.0371 | 0.1917 | 1.2183 | 0.1167 |
| Baseline + MMC | 44.86 | 25.27 | 0.0360 | 0.1875 | 1.2171 | 0.1076 |
| Baseline + MMC + SAC | 44.32 | 25.16 | 0.0362 | 0.1856 | 1.2106 | 0.1091 |
| Baseline + MMC + MAC (Ours) | **43.15** | **25.30** | **0.0355** | **0.1846** | **1.2039** | **0.1071** |

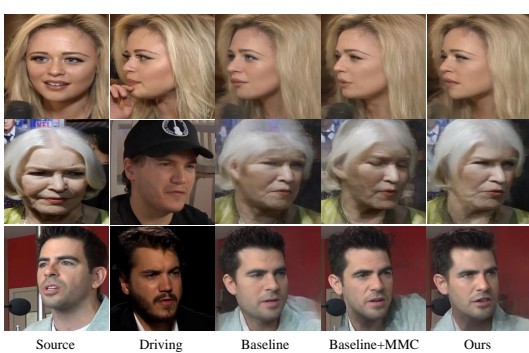

Figure 7: Visulization of the original and reconstructed motion flows for different scales. Our multi-scale motion codebook can reconstruct multi-scale motion flows with high quality.

### 4.3 ABLATION STUDY

We perform ablation studies to assess the effectiveness of the learned multi-scale motion and appearance compensatory codebooks. The model variants in Tab. 2 are as follows: **(i)** "Baseline" is the model without any compensatory codebook. **(ii)** "Baseline+SMC" includes only a single-scale motion codebook, compensating the initial motion flow at scale 1, which is used to warp multi-scale features. **(iii)** "Baseline+MMC" includes a multi-scale motion codebook to compensate the motion flow across scales. **(iv)** "Baseline+MMC+SAC" adds a single-scale appearance codebook to "Baseline+MMC," compensating only the warped feature at scale 1. **(v)** "Baseline+MMC+MAC" is the full model with both multi-scale motion and appearance codebooks. We present the quantitative results in Tab. 2 and qualitative comparisons of (i), (iii), and (v) in Fig. 9.

**Effect of Joint Learning of Multi-scale Motion and Appearance Codebooks.** To evaluate the effectiveness of our jointly learned multi-scale motion and appearance codebooks, we visualize the reconstructed multi-scale motion flows from the motion codebook in Fig. 7 and appearance features from the appearance codebook in Fig. 8, using vector quantization. For the motion flows, we use a motion flow decoder $\mathcal{D}_M$ to decode the quantized motion flow features and visualize the results. In Fig. 7, despite the limited number of codes, our multi-scale motion codebook can reconstruct high-quality motion flows, showing its ability to capture typical local motion patterns. For appearance features, we visualize the quantized features directly in Fig. 8. Despite some quantiza-

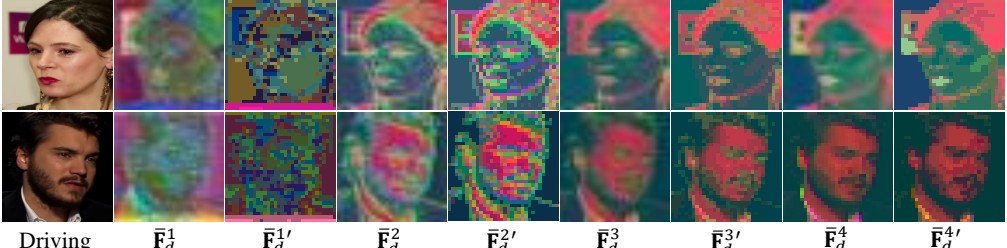

Figure 9: Qualitative ablation study on multi-scale motion and appearance codebook compensation. Both motion and appearance codebook compensation contribute to better generation quality.

tion loss, the multi-scale appearance codebook reconstructs the driving features well with limited codes, demonstrating the codebook's ability of storing informative local appearance details. Note that quantization errors are more noticeable at the feature level, and an image decoder specializing

Figure 8: Visulization of the original and reconstructed driving features. Our multi-scale appearance codebook can reconstruct multi-scale appearance features with acceptable quantization loss.

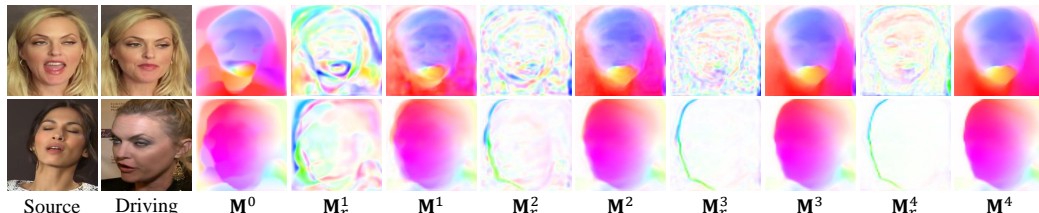

Figure 10: Visulization of the motion flow compensation process. We present the initial motion flow $\mathbf{M}^0$, the motion flow residual $\{\mathbf{M}_r^i\}_{i=1}^N$ and the compensated motion flows $\{\mathbf{M}^i\}_{i=1}^N$ for all scales.

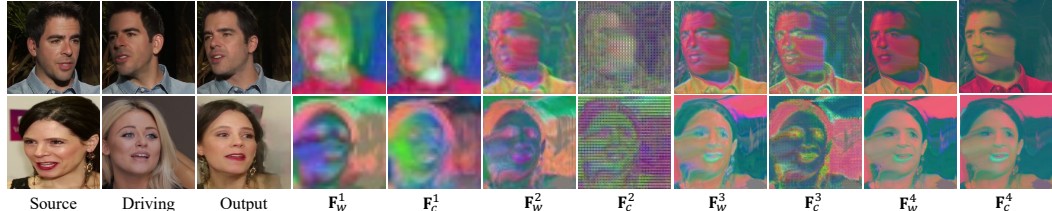

Figure 11: Visualization of the appearance compensation results. We present the warped source features $\{\mathbf{F}_w^i\}_{i=1}^N$ and the compensated appearance features $\{\mathbf{F}_c^i\}_{i=1}^N$ for all scales.

in decoding such quantized features can reduce the perceptual loss at the image level. These results demonstrate the effectiveness of jointly learning multi-scale motion and appearance codebooks, allowing them to store expressive local motion and appearance patterns for compensation.

**Effect of Multi-scale Motion Codebook Compensation.** The second row of Tab. 2 shows that using single-scale motion codebook compensation already improves motion transfer and image quality, reflected by better PSNR, $\mathcal{L}_1$, LPIPS, and AKD compared to the baseline. This highlights the effectiveness of motion codebook compensation for talking head video generation. Multi-scale motion codebook compensation further enhances all metrics, underscoring the importance of handling motion flows at different scales for improved feature warping. Fig. 9 shows that multi-scale motion compensation achieves better motion transfer (e.g., head pose in row 1), reduces artifacts (e.g., hair in row 2), and preserves source identity (row 3). Fig. 10 visualizes the motion flow compensation process, showing that the initial motion flow $\mathbf{M}^0$ is rough, but multi-scale compensation refines it iteratively with residuals $\{\mathbf{M}_r^i\}_{i=1}^N$, adding finer details as the scale increases for smoother, more face-adapted motion flows.

**Effect of Multi-scale Appearance Codebook Compensation.** The fourth row in Tab. 2 shows that adding single-scale appearance codebook compensation on top of multi-scale motion codebook compensation improves FID, LPIPS, and AKD, indicating that appearance codebook compensation refines warped source features for more realistic image generation. However, there is a slight drop in PSNR, $\mathcal{L}_1$, and AED, likely due to a conflict between scale 1 compensated appearance features and other warped features with warping artifacts. The fifth row shows consistent improvement, suggesting multi-scale appearance codebook compensation resolves this conflict and further refines warped features across scales, boosting overall performance. The last two columns in Fig. 9 also verify that multi-scale appearance codebook compensation leads to more accurate motion (e.g., the mouth in row 1, eyes in row 2, and shoulders in row 3) with realistic facial details (e.g., the hair in row 2). Additionally, Fig. 11 visualizes the compensated feature maps at different scales, showing more complete facial shapes and details compared to the warped feature $\mathbf{F}_w^i$. These results validate the effectiveness of multi-scale appearance codebook compensation.

## 5 CONCLUSION

In this paper, we present a novel framework that jointly learns multi-scale motion and appearance compensatory codebooks to enhance the motion flows and appearance features for talking head video generation. The jointly learned motion and appearance codebooks store local motion and appearance patterns learned from the entire dataset at different scales, and our multi-scale motion and appearance codebook compensation module retrieves useful codes from the codebooks with a transformer-based strategy at different scales to gradually refine the motion flows and appearance features during generation. Extensive results demonstrate the effectiveness of our codebook learning and compensation, synergizing motion and appearance to produce higher quality videos.

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
