# OpenReview forum: "Synergizing Motion and Appearance: Multi-Scale Compensatory Codebooks for Talking Head Video Generation"
_ICLR.cc/2025/Conference — ICLR 2025 Conference Withdrawn Submission_

### Official Review · Reviewer_rK7H · 2024-10-23

**Soundness:** 2
**Presentation:** 2
**Contribution:** 3
**Rating:** 6
**Confidence:** 4

**Summary:**

The authors propose a novel framework for talking head video generation, with the motivation of improving the capture of both the motion and appearance features. Towards this, the authors simultaneously learn multi-scale codebooks for both appearance and motion. The authors claim that the use of the proposed multi-scale refinement scheme at various granularities improves the visual quality of the generated videos. During inference, an initial motion flow estimate of the driving frame in reference to the source frame is refined progressively at various resolutions using features obtained by querying the motion codebook, which is used to warp the source facial features at each scale. These warped appearance features are further refined using the features obtained from the appearance codebook during the decoding process. The authors demonstrate superior results within the training distribution (VoxCeleb1) while yielding comparable results when evaluated on out-of-distribution samples (CelebV-HQ).

Note: I will be using a numbered list in the following sections, for ease of reference during the discussion phase.

**Strengths:**

1. Simultaneous refinement of both motion and appearance employing dedicated codebooks is a novel approach for talking-head generation.

2. The strategy of using multiple granularities to improve the generation quality especially when supplementing the decoding process with the appearance features is an interesting and intuitive approach.

3. The thinking behind most design choices has been explained.

4. The quantitative and qualitative results are promising, and the authors have compared their method across a multitude of baselines.

**Weaknesses:**

1. While it is true that cross-identity re-enactment does not have a groundtruth, a quantitative analysis based on metrics such as FVD or FID (preferably FVD as it models spatiotemporal correspondence), identity loss (face identity consistency)/CSIM, ARD, AU-H that have been used in prior research [A-D] would be expected.

2. While the qualitative results shown are promising in comparison to the baselines provided, there exist visible artifacts such as entanglements with the background (3rd cross-id video in supplementary), warped head shape in cross-id reenactment (eg: 2nd and 3rd cross-id video in supplementary), etc. While these limitations are reasonable given the complexity of the problem, discussing them in a limitations section is recommended.

3. The paper has a few clarity issues which I have highlighted in the section below.





[A] Guo, Jianzhu, et al. "Liveportrait: Efficient portrait animation with stitching and retargeting control." arXiv preprint arXiv:2407.03168 (2024).

[B] Ma, Yue, et al. "Follow-Your-Emoji: Fine-Controllable and Expressive Freestyle Portrait Animation." arXiv preprint arXiv:2406.01900 (2024).

[C] Yin, Fei, et al. "Styleheat: One-shot high-resolution editable talking face generation via pre-trained stylegan." European conference on computer vision. Cham: Springer Nature Switzerland, 2022.

[D] Doukas, Michail Christos, et al. "Free-headgan: Neural talking head synthesis with explicit gaze control." IEEE Transactions on Pattern Analysis and Machine Intelligence 45.8 (2023): 9743-9756.

**Questions:**

1. Support claims with examples (eg: L043.5 give examples of "motion patterns" that you are referring to. For example motion patterns such as near profile head motion)

2. In L47-48, the authors raise the limitation "still source image lacks sufficient appearance information to handle occluded regions or subtle expression changes". It is not clear how the proposed method overcomes this limitation. Is the codebook capable of hallucinating plausible facial features based on the query from the 'single source frame'? If so what prevents the model from generating unwarranted artifacts similar to the approaches that use generative models?

3. Could the authors clarify with reference to what the residual computed when they state "we predict the motion flow residual for each input" in L244? If it is the motion residual between successive frames, explaining what prevents error propagation and any measure taken to enforce temporal coherence would help the readers understand the process better since this is a video generation pipeline. I believe discussing these aspects in the method section in a "video" generative pipeline is important.

4. Is the pipeline robust to the initial keypoint-based coarse flow estimation? Is the choice of a keypoint based flow estimator random or a motivated?

5. Could the authors discuss the joint training of the codebooks (motion+appearance) compared to training each separately?

6. Minor typo in Figure 8: Visulization -> Visualization

---

### Official Review · Reviewer_wG6F · 2024-10-29

**Soundness:** 4
**Presentation:** 3
**Contribution:** 2
**Rating:** 5
**Confidence:** 4

**Summary:**

The article proposes to present jointly learned motion and appearance codebooks and multiscale motion and appearance compensation modules to enhance motion flow and appearance features generated from talking head videos. Specifically, the article designs a multi-scale motion and appearance codebook for storing representative global facial motion flow and appearance patterns. The multi-scale motion and appearance compensation module utilizes a transformer-based codebook retrieval strategy to query the complementary information of joint motion and appearance compensation from two codebooks. The whole framework effectively refines the motion conditions and appearance features of face images to generate high-quality speaker videos.

**Strengths:**

1. the method section of the article is described in detail and specifically, with diagrams and formulas for each sub-module.

2. The article is comprehensive in experiments and diverse in presentation. In addition to the comparative experiments on the topic, the article also shows many visualization results. The ablation experiment section verifies the effectiveness of the modules and strategies from multiple perspectives.

**Weaknesses:**

1. Insufficient Explanability. The introduction of the article explains the reasons for using the appearance compensation module but lacks an explanation of the reasons for using codebook learning, adding relevant explanations would help readers better understand the motivation and rationale behind choosing codebook learning as a solution to the problem of accurate pose and fine-grained facial details in Talking-head generation.

2. The methodology lacks innovation. The core modules, MCC and ACC, exhibit a structure akin to that presented in reference [1], and the approach for optimizing codebook learning remains unchanged. The paper merely applies these methods across multiple scales.

[1] Zhou S, Chan K, Li C, et al. Towards robust blind face restoration with codebook lookup transformer[J]. Advances in Neural Information Processing Systems, 2022, 35: 30599-30611.

**Questions:**

1. The method in this paper is trained on VoxCeleb1 and tested on VoxCeleb1 and CelebV-HQ. Is the same training set adopted for the other methods in the comparison method? The author is expected to give a detailed explanation and explanation of the training data set for each comparison method by providing a table or explicit statement in the paper detailing the exact training datasets used for each compared method.

---

### Official Review · Reviewer_HbAk · 2024-10-31

**Soundness:** 3
**Presentation:** 3
**Contribution:** 2
**Rating:** 5
**Confidence:** 5

**Summary:**

This paper proposed a new method for talking head generation. Two key component in the proposed method is (a) multi-scale motion and appearance codebooks which encodes representative motion and appearance patterns, and (b) a transformer-based compensation module which utilizes the two codebooks to compensation key-point based flow-maps as well as appearance features. Experiments on two talking head datasets validates the proposed method and demonstrates comparison with state-of-the-art talking head generation methods.

**Strengths:**

- The idea of using multi-scale feature for synthesis makes sense.

- The overall writing is clear and easy to follow.

- Extensive ablation studies for validation the proposed components.

**Weaknesses:**

(1) The novelty of using multi-scale features. The usage of mutli-scale features are quite straightforward in many computer vision applications. Regarding talking head generation, previous works such as [1] has proposed to use multi-scale appearance features to form a tri-plane for talking head generation. Despite of different final representations (implicit field v.s. key-point based), the idea of introducing multi-scale features is similar and there seems no discussions in this paper.

(2) The motivation of introducing codebooks for compensate motion and appearance is not clear enough. A straight forward method, at first glance, would be directly output residuals for compensation without using codebooks. One guess (from my POV) is that the learned codebooks constrained the operation space when performing compensations with input flow maps / appearance features, preventing arbitrary residuals which may destroy the output. However, this is purely my guess and there are no discussions of the motivations behind the usage of these codebooks. It would be helpful if the paper could explain why codebooks are preferable to directly outputting residuals, and what specific advantages the proposed method offers for talking head generation.

(3) The result quality has not significantly outperform previous works. Per table 1, the proposed method does not outperform other baselines on CelebV-HQ dataset quantitatively. In figure 5, one could easily observe that the synthesized image has unconvincing identity preservation when performing cross-identity reenactments (e.g., in the first row and the second row the generated image has a wider face that similar to the driving person instead of the source; in the last row the generated image has a round chin while the source image has a pointed chin). Overall, these results raise my concerns that whether the current result is good enough to validate the effectiveness of the proposed method. Providing a more in-depth analysis of the method's performance particularly on the CelebV-HQ dataset where it failed to outperform other baselines would be helpful.

References:

[1] Li, Weichuang, et al. "One-shot high-fidelity talking-head synthesis with deformable neural radiance field." Proceedings of the IEEE/CVF Conference on Computer Vision and Pattern Recognition. 2023.

**Questions:**

The transformer blocks used for compensate motion flow and appearance features could be regarded as a "implicit warping" operation, similar to [1]. In [1], the implicit warping approach was proposed to utilize multiple reference images, while this paper argues a multi-scale approach with one-shot reference. As the paper mentioned in the introduction, two key challenges are "(1) Some motion patterns cannot be inferred from a single image pair, and (2) a still source image lacks sufficient appearance information to handle occlude regions or subtle expression changes". Why does the proposed multi-scale approach could address these challenges with a single image input? Could you provide some more insights behind it? Will multiple reference images helpful?

References:

[1] Mallya, Arun, Ting-Chun Wang, and Ming-Yu Liu. "Implicit warping for animation with image sets." Advances in Neural Information Processing Systems 35 (2022): 22438-22450.

---

### Official Review · Reviewer_vU2M · 2024-11-02

**Soundness:** 2
**Presentation:** 2
**Contribution:** 2
**Rating:** 5
**Confidence:** 4

**Summary:**

This paper argues that currently, under one shot talking head video generation (video driven setting), the learning of motion and appearance remains a difficult problem to solve. It proposes a multiscale codebook structure to enhance the synthesis model from the aspects of motion and appearance. Considering the provided demos and comparison experiments, this article has achieved relatively good results.

**Strengths:**

(1) Using the codebook can increase the model's expression capacity for motion and appearance features within the existing framework. This is meaningful.

(2) Looking at the experimental results, the videos generated by the method in this article have advantages in some aspects compared to existing methods, such as clarity.

(3) The experiments are relatively complete.

**Weaknesses:**

(1) When we check out the experimental results, we see that the method in this paper isn't much better in terms of indicators. For example, on the VoxCeleb1 dataset, for PSNR and AKD, the baseline (Table 2) is already almost as good as or even better than TPSM and MCNet (Table 1). Looks like this paper has a pretty strong baseline.

(2) The experimental results can't really show that the codebook works well in both appearance and motion. From the visualized results, the method in this paper does well in clarity compared to existing ones. But from Table 2, it seems that MMC is more important for the improvement in PSNR and FID. The result of MAC isn't obvious. For motion, MMC can make some improvement, but it's not much better than other methods (like having a worse AKD than MRFA under same-identity reconstruction).

**Questions:**

(1) Looking at Figure 9, both MAC and MCC have nice improvements at the frame level. But the author should give multi-frame results or video results. It is not certain how much difference there is in the overall generated video. That would better show the differences in effects and include more test cases, such as 1) gradual pose changes from frontal face to side face and 2) expression transitions from plain to extreme.

(2) How much computational cost (FLOPs) and parameter size will be increased by the addition of MAC and MMC? Or in terms of running time, there should be some explanations here. More specifically, it can include the following contents: 1) computational costs for each component (MAC and MMC) separately, 2) as well as the overall increase compared to the baseline model; 3)The running time of different methods on the same device.


(3) In the test examples, most of the motions are still the absolute motion of the head plus simple expression motions. For methods like follow-your-emoji and liveportrait, they have good performance in controlling subtle expressions[1-3]. How does the current method perform in this aspect? And what are the effect advantages compared with the comparison method? For example, the cases in links [4].



In general, I think the results obtained in this article are not bad. However, experimentally, it is not well supported that the codebook can improve motion and appearance at the same time. It seems that more often, the introduction of MMC will improve the final clarity to a certain extent.



[1] https://liveportrait.github.io/src/video/image_animation_0_opt.mp4

[2] https://www.youtube.com/watch?v=kg9qpyupXbI#t=17

[3] https://byteaigc.github.io/x-portrait/

[4] https://kwaivgi-liveportrait.hf.space/file=/tmp/gradio/34db59a6dca64fde3f059ebeb888dd31a71e5592/d6_trim.mp4

---

### Note · Authors · 2024-11-15

I have read and agree with the venue's withdrawal policy on behalf of myself and my co-authors.